# Antenna Pattern Calibration Method for Phased Array of High-Frequency Surface Wave Radar Based on First-Order Sea Clutter

**Hongbo Li** [1] , **Aijun Liu** [2,*] , **Qiang Yang** [1] , **Changjun Yu** [2] **and Zhe Lyv** [2]

1  School of Information and Electrical, Harbin Institute of Technology, Harbin 150000, China; 19B905046@stu.hit.edu.cn (H.L.); yq@hit.edu.cn (Q.Y.)
2  School of Information and Electrical, Harbin Institute of Technology (Weihai), Weihai 264200, China; yuchangjun@hit.edu.cn (C.Y.); lvzhehit@hit.edu.cn (Z.L.)
*  Correspondence: liuaijun@hit.edu.cn

**Abstract:** The problem of accurate source localization has been an area of focus in high-frequency surface wave radar (HFSWR) applications. However, antenna pattern distortion (APD) decreases the direction-of-arrival (DOA) estimation performance of the multiple signal classification (MUSIC) algorithm. Up to now, limited studies have been conducted on the calibration of antenna pattern distortion for phased arrays in HFSWR. In this paper, we first analyze the effect of APD on the performance of the MUSIC algorithm through estimation of accuracy and angular resolution. We demonstrate that using the actual pattern (or say APD) can improve DOA estimation performance. Based on this proposition, we propose a novel iterative calibration method that employs the first-order sea clutter data and can jointly estimate DOA and APD in an iterative way. To obtain available calibration points, we introduce the extraction methods of the first-order sea clutter spectrum and single-DOA spectrum points. Meanwhile, in each iteration, the Beamspace MUSIC algorithm and artificial hummingbird algorithm (AHA) are utilized to estimate the DOA and APD, respectively. Numerical results reveal a good coincidence between the actual pattern and the estimated APD. We also apply this method to process the experimental data of HFSWR. We obtain the APD vector of the real phased array and improve the direction-finding performance of several real ship targets using this vector. Both numerical and experimental results prove the correctness of our proposed calibration method.

**Keywords:** high-frequency surface wave radar; phased array; antenna pattern distortion; first-order sea clutter; single-DOA spectrum point; beamspace multiple signal classification algorithm; artificial hummingbird algorithm



## 1. Introduction

Over the past few decades, high-frequency surface wave radar (HFSWR) has become one of the most widely used and advanced next-generation ocean detection technologies due to its extensive working range and strong real-time characteristics. It operates in the bandwidth of 3–30 MHz and can detect ships, icebergs, missiles and other moving targets at sea at distances of more than 300 km. In addition, the detection results are provided in real time at various time resolutions (from minutes to tens of minutes) and spatial resolutions (from hundreds of meters to kilometers). These results have attracted significant attention and played essential roles in many civil and military applications, including remote sensing [1], ocean surface current [2], wind measurement [3] and ship detection [4].

In these applications, accurate source localization has long been of significance in research, given their importance in solving several real-world problems. To achieve this goal, two types of methods are commonly used for direction-of-arrival (DOA) estimation. One method is beam forming (BF), which is represented by the conventional beamforming

(CBF) algorithm [5]. The other method is direction finding (DF), which is represented by the multiple signal classification (MUSIC) algorithm [6]. Although their principles differ, both types of methods rely on a preknown received array model and an ideal received signal model to estimate DOA. This means that these methods suffer from a fundamental limitation: their performances are susceptible to array uncertainties and degrade significantly once the models deviate. With the existence of deviations, the BF method generally obtains incorrect radiation patterns with broadened beam width and high side lobes [7], and the DF method yields spurious directions and poor angular resolution [8, 9]. In fact, the deviations (errors) are inevitable, so error calibration is necessary.

Generally, the errors are divided into four kinds: gain-phase error (GPE) [10], array position error (APE) [11], mutual coupling error (MCE) [12] and antenna pattern distortion (APD) [13]. Research on the calibration algorithms of the first three errors is relatively mature due to their simple error models, which are defined as independent of the angle of the incident signal. These algorithms are mainly divided into two categories: active calibration algorithms [14,15] and passive calibration algorithms [16,17]. Active calibration algorithms estimate the error parameters offline by setting an auxiliary source with an exactly known azimuth in space. They have an excellent calibration effect but require high-quality auxiliary sources, which means an increase in the cost of direction-finding systems. Passive calibration algorithms construct a particular cost function based on principles, through which the azimuth of the opportunity sources and the error parameters of the array are estimated jointly. Although they do not require an auxiliary source, high-SNR opportunity sources are not always available.

The error model of the APD is quite complicated, and its error parameters vary with the incident angle. This indicates that a single auxiliary source (for active calibration) or several opportunity sources (for passive calibration) cannot calibrate errors in all directions. Some studies have suggested that measuring the actual antenna pattern directly instead of calibrating the APD [18–20]. To obtain the measured pattern, these studies apply a ship carrying a transponder to move along a predetermined path. However, these measurements are costly and not always satisfactory, depending on the weather and topography of the coast. Although an aerial drone has been designed as a carrier for the transponder to reduce the cost, this method is limited by the endurance of the drone [21]. Meanwhile, similar to active calibration, some methods have combined ship echoes and automatic identification system (AIS) data to estimate the antenna pattern [22–24]. These methods only involve numerical calculations and do not require an actual transponder. However, to cover all interested angles, they require a large amount of ship echoes, which usually take several days or longer to record. Note that the sea echoes already contain signals from all directions, so some antenna pattern estimation/calibration methods using sea echoes have been proposed [25–28]. As a kind of passive calibration, these methods employ different iterative algorithms and cost functions to calibrate the APD. They can achieve real-time and automatic calibration without costly experiments or prolonged recordings. The drawback is that these methods are only suited to cross-loop/monopole antenna arrays with special structures, with a lack of discussions of ordinary phased arrays or arbitrary arrays.

In this paper, we propose a novel iterative calibration method for the antenna pattern distortion of high-frequency surface wave radar. The calibration method employs first-order sea clutter data as the calibration source and can realize the joint estimation of DOA and APD through iteration. We first analyze the performance of the MUSIC algorithm for a phased array and derive explicit expressions for its estimation accuracy and angular resolution under the ideal pattern and APD. Through theoretical and numerical comparisons, we demonstrate that the use of the actual pattern can improve the performance of DOA estimation. Then, we propose an iterative calibration method for the APD of HFSWR. In each iteration, the Beamspace MUSIC (BMUSIC) algorithm and artificial hummingbird algorithm (AHA) are utilized to estimate the DOA and APD, respectively. Meanwhile, to secure valid calibration sources, we introduce the extraction methods of first-order sea clutter spectrum and single-DOA spectrum points to preprocess the sea clutter data.

Numerical and experimental results verify the reliability of our proposed method. We obtain a good coincidence between the actual pattern and estimated APD (especially in the $[-30°, 30°]$ angle range) and enhance the direction-finding performance of several real ship echoes using the estimated APD. By averaging, we obtain an amplitude improvement of about 10 dB and an accuracy improvement of about $2°$.

## 2. Direction-Finding Problem Formulation

In this section, we introduce the received signal model of a phased array with the ideal antenna pattern and antenna pattern distortion. We employ the MUSIC algorithm for direction finding and analyze the effect of APD on DOA estimation.

### 2.1. Signal Model

For a common HFSWR phased array, we consider the following signal assumptions:

1. The sources are narrow-band signals and conform to the far-field point-source model;
2. The sources are uncorrelated with each other;
3. The noises are additive white Gaussian noise (AWGN) and uncorrelated with the sources.

Under the above assumptions, the received signal vector of the phased array is

$$\mathbf{X}(t) = \mathbf{A}(\theta)\mathbf{S}(t) + \mathbf{N}(t) \tag{1}$$

where

- $\mathbf{X}(t) = [x_1(t), x_2(t), \cdots, x_M(t)]^{\mathrm{T}}$ is the received signal vector, with $M$ as the number of antennas; and the superscript T denotes the transposition;
- $\mathbf{S}(t) = [s_1(t), s_2(t), \cdots, s_N(t)]^{\mathrm{T}}$ is the incoming signal vector, with $N$ as the number of sources;
- $\mathbf{A}(\theta) = [\mathbf{a}(\theta_1), \mathbf{a}(\theta_2), \cdots, \mathbf{a}(\theta_N)]$ is the array manifold matrix, with $\theta_n$ as the azimuth of the $n$-th source;
- $\mathbf{a}(\theta_n) = [\exp(-j\omega\tau_1^n), \exp(-j\omega\tau_2^n), \cdots, \exp(-j\omega\tau_M^n)]^{\mathrm{T}}$ for the array, with $\omega$ as the radar operating frequency;
- $\tau_m^n = (m-1)d\sin\theta_n/c$ for a uniform linear array (ULA), with $d$ and $c$ representing the antenna spacing and speed of light, respectively;
- $\mathbf{N}(t)$ is the AWGN with zero mean and variance ($\sigma^2$).

Note that Equation (1) does not involve the antenna pattern, implying that it represents the received signal under the ideal pattern. At the actual array site, the antenna pattern will be distorted by the surrounding obstacles, such as high buildings, massifs and trees. Without loss of generality, the APD is defined as the ratio of the actual pattern to the ideal pattern

$$f_m(\theta) = g_m(\theta)\exp(j\varphi_m(\theta)) \tag{2}$$

where $m = 1, 2, \cdots, M$ represents the index of the antenna elements, and $g(\theta)$ and $\varphi(\theta)$ represent the gain distortion and phase distortion, respectively. The first antenna is usually set as the reference antenna, so $f_1(\theta) = 1$. In addition, this distortion model should already include the GPE and APE, since they only impose a gain-phase error constant on the array.

In the case in which the array has APDs, the array manifold should be modified as

$$\mathbf{a}_{\mathrm{APD}}(\theta) = [f_1(\theta)\exp(-j\omega\tau_1), f_2(\theta)\exp(-j\omega\tau_2), \cdots, f_M(\theta)\exp(-j\omega\tau_M)]^{\mathrm{T}} \tag{3}$$

It can be written more concisely as

$$\mathbf{a}_{\mathrm{APD}}(\theta) = \mathbf{f}(\theta) \odot \mathbf{a}(\theta) \tag{4}$$

where $\mathbf{f}(\theta) = [f_1(\theta), f_2(\theta), \cdots, f_M(\theta)]^{\mathrm{T}}$ and $\odot$ denotes the Hadamard product.

Then, the received signal model represented in Equation (1) should also be rewritten using the modified array manifold as

$$\mathbf{X}_{\text{APD}}(t) = \mathbf{A}_{\text{APD}}(\theta)\mathbf{S}(t) + \mathbf{N}(t) \tag{5}$$

where $\mathbf{A}_{\text{APD}}(\theta) = [\mathbf{f}(\theta_1) \odot \mathbf{a}(\theta_1), \mathbf{f}(\theta_2) \odot \mathbf{a}(\theta_2), \cdots, \mathbf{f}(\theta_N) \odot \mathbf{a}(\theta_N)]$.

*2.2. MUSIC Algorithm Model*

Due to its super-resolution, the MUSIC algorithm has been the most commonly used algorithm in source localization. To achieve DOA estimation of the sources, the MUSIC algorithm first constructs the covariance matrix of the measurements

$$\mathbf{R}_{\text{X}} = E\left[\mathbf{X}(t)\mathbf{X}^{\text{H}}(t)\right] = \mathbf{A}\mathbf{R}_{\text{S}}\mathbf{A}^{\text{H}} + \mathbf{R}_{\text{N}} = \mathbf{A}\mathbf{R}_{\text{S}}\mathbf{A}^{\text{H}} + \sigma^2\mathbf{I} \tag{6}$$

where $\mathbf{A}$ is a brief notation for $\mathbf{A}(\theta)$; the superscript H denotes the Hermitian transpose; $\mathbf{I}$ is the unit matrix; and $\mathbf{R}_{\text{S}}$ and $\mathbf{R}_{\text{N}}$ represent the signal covariance matrix and noise covariance matrix, respectively. In practice, the covariance matrix is consistently estimated by the actual data, i.e., $(1/L)\sum_{t=1}^{L}\mathbf{X}(t)\mathbf{X}^{\text{H}}(t)$, where $L$ denotes the number of snapshots.

Then, by carrying out an eigenvalue decomposition on the covariance matrix ($\mathbf{R}_{\text{X}}$), the noise subspace of the measurements is determined

$$\mathbf{R}_{\text{X}} = \sum_{k=1}^{M} \lambda_k \mathbf{u}_k \mathbf{u}_k^{\text{H}} = \mathbf{U}_{\text{S}}\Sigma_{\text{S}}\mathbf{U}_{\text{S}}^{\text{H}} + \mathbf{U}_{\text{N}}\Sigma_{\text{N}}\mathbf{U}_{\text{N}}^{\text{H}} \tag{7}$$

where

- $\lambda_1 \geq \lambda_2 \geq \cdots \geq \lambda_N > \lambda_{N+1} = \cdots = \lambda_M = \sigma^2$ represents the eigenvalues of $\mathbf{R}_{\text{X}}$, and the first $N$ large eigenvalues are generated by the sources, while the last $M-N$ small eigenvalues are caused by the noises;
- $u_k$ denotes the eigenvector corresponding to the $k$-th eigenvalue;
- $\mathbf{U}_{\text{S}} = [\mathbf{u}_1, \cdots, \mathbf{u}_N]$ and $\mathbf{U}_{\text{N}} = [\mathbf{u}_{N+1}, \cdots, \mathbf{u}_M]$ represent the signal subspace and noise subspace, respectively, and they are orthogonal complement spaces.

Finally, the MUSIC algorithm estimates DOA by minimizing the spectrum function, which is defined as the projection of the array manifold on the noise subspace:

$$P(\theta) = \mathbf{a}^{\text{H}}(\theta)\mathbf{U}_{\text{N}}\mathbf{U}_{\text{N}}^{\text{H}}\mathbf{a}(\theta) \tag{8}$$

Note that with the ideal pattern, the signal subspace and array manifold matrix should span the same subspace, i.e., span$\{\mathbf{A}(\theta)\}$ = span$\{\mathbf{U}_{\text{S}}\}$. In this case, $\mathbf{A}(\theta)$ is orthogonal to the noise subspace ($\mathbf{U}_{\text{N}}$), and Equation (8) takes the minimum values at $\theta = \theta_1, \cdots, \theta_N$. However, with the APD, the signal subspace should span the same subspace as the actual array manifold matrix ($\mathbf{A}_{\text{APD}}(\theta)$). Thus, the orthogonality of $\mathbf{A}(\theta)$ and $\mathbf{U}_{\text{N}}$ is reduced, and the DOAs estimated by Equation (8) deviate from the actual incident angles. In the next section, we analyze the effect of APD on the performance of MUSIC in detail.

## 3. Performance of the MUSIC Algorithm with Antenna Pattern Distortion

In this section, we analyze the effect of APD on the performance of the MUSIC algorithm in terms of estimation accuracy and angular resolution. We demonstrate that the performance can be improved by using the actual measured pattern to estimate DOA.

*3.1. Estimation Accuracy of the MUSIC Algorithm*

Although the estimation accuracy of MUSIC was evaluated with respect to the error variance indicator in [29], the research mainly considered the ideal pattern. According to [29], the error variances of the uncorrelated sources are given by

$$E\left(\hat{\theta}_i - \theta_i\right)^2 = \frac{1}{2L}\frac{\mathbf{a}^{\text{H}}(\theta_i)\mathbf{U}\mathbf{a}(\theta_i)}{h(\theta_i)} \tag{9}$$

where $\hat{\theta}_i$ and $\theta_i$ denote the estimated and actual DOA of the $i$-th source, respectively, and

$$\mathbf{U} = \sum_{k=1}^{N} \frac{\lambda_k \sigma^2}{(\lambda_k - \sigma^2)^2} \mathbf{u}_k \mathbf{u}_k^{\mathrm{H}} \tag{10}$$

$$h(\theta) = \mathbf{d}^{\mathrm{H}}(\theta) \left( \mathbf{I} - \mathbf{U}_{\mathrm{S}} \mathbf{U}_{\mathrm{S}}^{\mathrm{H}} \right) \mathbf{d}(\theta) \tag{11}$$

where $\mathbf{d}(\theta) = d\mathbf{a}(\theta)/d\theta$ is the derivative of the array manifold with respect to $\theta$.

Notably, Equation (9) only involves the case of the ideal pattern. When considering actual applications, it should be extended to the following three cases (see [9]) due to the inevitable APD. To simplify the derivation, we assume a single-source environment in the following analyses, that is, $N = 1$.

*Case 1 (Ideal Pattern and MUSIC Estimation with Ideal Pattern):* This is exactly the case represented by Equation (9), and we take it as a reference. For a phased array composed of $M$ dipole antennas, we repeat its array manifold with the ideal pattern as

$$\mathbf{a}(\theta) = [\exp(-j\omega\tau_1), \exp(-j\omega\tau_2), \cdots, \exp(-j\omega\tau_M)]^{\mathrm{T}}, \tag{12}$$

and its derivative $\mathbf{d}(\theta)$ is

$$\mathbf{d}(\theta) = \left[ -j\omega\tau_1' \exp(-j\omega\tau_1), -j\omega\tau_2' \exp(-j\omega\tau_2), \cdots, -j\omega\tau_M' \exp(-j\omega\tau_M) \right]^{\mathrm{T}} \tag{13}$$

We now derive the concrete expressions of $\mathbf{U}$ and $h(\theta)$. Under the single-source assumption, we have $\mathbf{U}_{\mathrm{S}} = \mathbf{u}_1$. Meanwhile, the signal subspace in this case spans the same subspace as the ideal array manifold (span$\{\mathbf{u}_1\} = $ span$\{\mathbf{a}(\theta)\}$). Since both $\mathbf{u}_1$ and $\mathbf{a}(\theta)/\|\mathbf{a}(\theta)\|$ are orthonormal bases of the spanned subspace, we have

$$\mathbf{U}_{\mathrm{S}} = \mathbf{u}_1 = \frac{\mathbf{a}(\theta)}{\|\mathbf{a}(\theta)\|} \tag{14}$$

By inserting Equation (14) into (10), we expand $\mathbf{U}$ as

$$\mathbf{U}^{(1)} = \left[ \frac{\sigma^2}{\lambda_1 - \sigma^2} + \frac{\sigma^4}{(\lambda_1 - \sigma^2)^2} \right] \frac{\mathbf{a}(\theta)\mathbf{a}^{\mathrm{H}}(\theta)}{\|\mathbf{a}(\theta)\|^2} = \left[ 1 + \frac{1}{\|\mathbf{a}(\theta)\|^2 \cdot \mathrm{SNR}} \right] \frac{\mathbf{a}(\theta)\mathbf{a}^{\mathrm{H}}(\theta)}{\|\mathbf{a}(\theta)\|^4 \cdot \mathrm{SNR}} \tag{15}$$

where $\lambda_1 = P\|\mathbf{a}(\theta)\|^2 + \sigma^2$ is used, with $P$ as the signal power and SNR as the signal-to-noise ratio (SNR) of the source.

We can also obtain $h(\theta)$ by inserting Equations (12), (13) and (14) into (11)

$$h^{(1)}(\theta) = \sum_{m=1}^{M} \left( \omega\tau_m' \right)^2 - \frac{1}{M} \left( \sum_{m=1}^{M} \omega\tau_m' \right)^2 \tag{16}$$

Then, we obtain the error variance of MUSIC—in this case, as

$$\mathrm{var}^{(1)}(\hat{\theta}) = \frac{1}{2L \cdot \mathrm{SNR}} \left( 1 + \frac{1}{M \cdot \mathrm{SNR}} \right) \Big/ h^{(1)}(\theta) \tag{17}$$

*Case 2 (APD but MUSIC Estimation with Ideal Pattern):* This is a common case if error calibration is neglected, as distortion is inevitable. In this case, we still employ the ideal array manifold shown in Equation (12) to solve the error variance. The key distinction is that the covariance matrix is estimated by the actual received data with APD, so the signal subspace should span the same subspace as the actual array manifold (span$\{\mathbf{u}_1\} = $ span$\{\mathbf{a}_{\mathrm{APD}}(\theta)\}$). Then, we have

$$\mathbf{U}_{\mathrm{S}} = \mathbf{u}_1 = \frac{\mathbf{a}_{\mathrm{APD}}}{\|\mathbf{a}_{\mathrm{APD}}(\theta)\|}, \tag{18}$$

and the actual array manifold is

$$\mathbf{a}_{\text{APD}}(\theta) = \mathbf{f}(\theta) \odot \mathbf{a}(\theta) = [g_1 \exp(-j(\omega\tau_1 - \varphi_1)), g_2 \exp(-j(\omega\tau_2 - \varphi_2)), \cdots, g_M \exp(-j(\omega\tau_M - \varphi_M))]^{\text{T}} \tag{19}$$

where $g_m$ and $\varphi_m$ are the brief notations for $g_m(\theta)$ and $\varphi_m(\theta)$, respectively.

According to Equations (18) and (19), we have

$$\mathbf{U}^{(2)} = \left[1 + \frac{1}{\|\mathbf{a}_{\text{APD}}(\theta)\|^2 \cdot \text{SNR}}\right] \frac{\mathbf{a}_{\text{APD}}(\theta)\mathbf{a}_{\text{APD}}^{\text{H}}(\theta)}{\|\mathbf{a}_{\text{APD}}(\theta)\|^4 \cdot \text{SNR}} \tag{20}$$

where we implicitly use $\lambda_1 = P\|\mathbf{a}_{\text{APD}}(\theta)\|^2 + \sigma^2$.

Based on Equations (13), (18) and (19), we have

$$h^{(2)}(\theta) = \sum_{m=1}^{M} (\omega\tau_m)^2 - \frac{1}{\sum_{m=1}^{M} g_m^2} \left[ \left( \sum_{m=1}^{M} \omega\tau'_m g_m \cos\varphi_m \right)^2 + \left( \sum_{m=1}^{M} \omega\tau'_m g_m \sin\varphi_m \right)^2 \right] \tag{21}$$

Then, we can derive the error variance in this case as

$$\text{var}^{(2)}(\hat{\theta}) = \frac{1}{2L \cdot \text{SNR}} \left[ 1 + \frac{1}{\sum_{m=1}^{M} g_m^2 \cdot \text{SNR}} \right] \frac{\left(\sum_{m=1}^{M} g_m \cos\varphi_m\right)^2 + \left(\sum_{m=1}^{M} g_m \sin\varphi_m\right)^2}{\left(\sum_{m=1}^{M} g_m^2\right)^2 h^2(\theta)} \tag{22}$$

*Case 3 (APD and MUSIC Estimation with Actual Pattern):* In this case, the actual pattern is measured or the distortion is calibrated, and we utilize the actual manifold shown in Equation (19) to calculate the error variance. Therefore, the derivative of the array manifold is

$$\begin{aligned} \mathbf{d}(\theta) &= d[\mathbf{f}(\theta) \odot \mathbf{a}(\theta)]/d\theta = \mathbf{f}'(\theta) \odot \mathbf{a}(\theta) + \mathbf{f}(\theta) \odot \mathbf{a}'(\theta) \\ &= [\beta_1 \exp(-j(\omega\tau_1 - \varphi_1)), \beta_2 \exp(-j(\omega\tau_2 - \varphi_2)), \cdots, \beta_M \exp(-j(\omega\tau_M - \varphi_M))] \end{aligned} \tag{23}$$

where $\beta_m = g'_m - jg_m(\omega\tau'_m - \varphi'_m)$.

Meanwhile, the signal subspace should also align with the actual array manifold, so Equation (18) is also employed. Obviously, $\mathbf{U}$, in this case, is the same as in *Case 2*:

$$\mathbf{U}^{(3)} = \mathbf{U}^{(2)} \tag{24}$$

By substituting Equations (18), (19) and (23) into (11), we derive $h(\theta)$ as

$$h^{(3)}(\theta) = \sum_{m=1}^{M} \left[ g'^2_m + g_m^2(\omega\tau'_m - \varphi'_m)^2 \right] - \frac{1}{\sum_{m=1}^{M} g_m^2} \left[ \left( \sum_{m=1}^{M} g_m g'_m \right)^2 + \left( \sum_{m=1}^{M} g_m^2(\omega\tau'_m - \varphi'_m) \right)^2 \right] \tag{25}$$

Then, the error variance is

$$\text{var}^{(3)}(\hat{\theta}) = \frac{1}{2L \cdot \text{SNR}} \left( 1 + \frac{1}{\sum_{m=1}^{M} g_m^2 \cdot \text{SNR}} \right) \Big/ h^{(3)}(\theta) \tag{26}$$

### 3.2. Angular Resolution of the MUSIC Algorithm

Until now, there has been no consensus definition of the angular resolution of the MUSIC algorithm. In this paper, we apply the concept of zero spectrum (ZS) proposed in [30] to evaluate the angular resolution. For two sources with adjacent incident angles, the MUSIC algorithm can distinguish them if they satisfy

$$Z(\theta_{\text{m}}) = e(\theta_i) - e(\theta_m), \qquad i = 1, 2 \tag{27}$$

where

- $Z(\theta) = \mathbf{a}^{\text{H}}(\theta)\mathbf{U}_{\text{N}}\mathbf{U}_{\text{N}}^{\text{H}}\mathbf{a}(\theta)$ is the zero spectrum of the MUSIC algorithm;
- $\theta_1$ and $\theta_2$ are the incident angles of two sources, and $\theta_{\text{m}} = (\theta_1 + \theta_2)/2$ is their midpoint;

- $e(\theta) = (M-2)\omega_2\left|\mathbf{a}^{\mathrm{H}}(\theta)\mathbf{u}_2\right|^2\Big/L$, where $\omega_2 = \lambda_2\sigma^2\Big/\left(\lambda_2-\sigma^2\right)^2$.

The expanded form of Equation (27) is

$$\|\mathbf{a}(\theta_m)\|^2 - |\mathbf{a}^{\mathrm{H}}(\theta_m)\mathbf{u}_1|^2 - |\mathbf{a}^{\mathrm{H}}(\theta_m)\mathbf{u}_2|^2 > \frac{M-2}{L}\omega_2\left(|\mathbf{a}^{\mathrm{H}}(\theta_i)\mathbf{u}_2|^2 - |\mathbf{a}^{\mathrm{H}}(\theta_m)\mathbf{u}_1|^2\right) \quad (28)$$

Note that $\lambda_2$ corresponds to the source with lower SNR and is approximately expressed as:

$$\lambda_2 = P_2\|\mathbf{a}(\theta_m)\|^2\left[1 - \frac{|\mathbf{a}^{\mathrm{H}}(\theta_1)\mathbf{a}(\theta_2)|}{\|\mathbf{a}(\theta_1)\|\cdot\|\mathbf{a}(\theta_2)\|}\right] + \sigma^2 = \eta\cdot P_2 + \sigma^2 \quad (29)$$

where $\eta = \|\mathbf{a}(\theta_m)\|^2\left[1 - |\mathbf{a}^{\mathrm{H}}(\theta_1)\mathbf{a}(\theta_2)|/(\|\mathbf{a}(\theta_1)\|\cdot\|\mathbf{a}(\theta_2)\|)\right]$.

Therefore, we simplify $\omega_2$ as

$$\omega_2 = \frac{\sigma^2}{\lambda_2-\sigma^2} + \frac{\sigma^4}{(\lambda_2-\sigma^2)^2} = \frac{1}{\eta\cdot\mathrm{SNR}_2} + \frac{1}{(\eta\cdot\mathrm{SNR}_2)^2} \quad (30)$$

where $\mathrm{SNR}_2 = P_2/\sigma^2$ is the SNR of the weaker source.

Inserting Equation (30) into (28), we derive the final angular resolution criterion as

$$L > (M-2)\frac{|\mathbf{a}^{\mathrm{H}}(\theta_i)\mathbf{u}_2|^2 - |\mathbf{a}^{\mathrm{H}}(\theta_m)\mathbf{u}_1|^2}{\|\mathbf{a}(\theta_m)\|^2 - |\mathbf{a}^{\mathrm{H}}(\theta_m)\mathbf{u}_1|^2 - |\mathbf{a}^{\mathrm{H}}(\theta_m)\mathbf{u}_2|^2}\left[\frac{1}{\eta\cdot\mathrm{SNR}_2} + \frac{1}{(\eta\cdot\mathrm{SNR}_2)^2}\right] \quad (31)$$

It is observed that the angular resolution threshold is determined by several parameters, including the SNR of the weaker source, the number of snapshots and the angular difference between two sources. In the later discussion, we consider the angular resolution as the number of snapshots required to distinguish between two sources with a given angular difference.

In addition, similar to the analysis of the estimated accuracy, Equation (31) should also be discussed in three cases. To avoid redundancy, we report the selection of array manifolds in different cases and eigenvectors instead of the specific formulas.

*Case 1:* Select $\mathbf{a}(\theta)$ and $\mathbf{u}_i = \mathbf{a}(\theta_i)/\|\mathbf{a}(\theta_i)\|$.
*Case 2:* Select $\mathbf{a}(\theta)$ and $\mathbf{u}_i = \mathbf{a}_{\mathrm{APD}}(\theta_i)/\|\mathbf{a}_{\mathrm{APD}}(\theta_i)\|$.
*Case 3:* Select $\mathbf{a}_{\mathrm{APD}}(\theta)$ and $\mathbf{u}_i = \mathbf{a}_{\mathrm{APD}}(\theta_i)/\|\mathbf{a}_{\mathrm{APD}}(\theta_i)\|$.

### 3.3. Comparison of the MUSIC Algorithm's Performance

Since the APD destroys the orthogonality of the ideal array manifold and the actual noise subspace, there is no doubt that the DOA estimation performance in *Case 2* is worse than that in *Case 1*. The focus of the comparison is to explore the relationship between DOA estimation performance in *Case 1* and *Case 3*. In fact, the APD can be divided into three kinds: only phase-pattern distortion exists, only gain-pattern distortion exists or both phase- and gain-pattern distortion exist. Due to the complicated nature of the formulas, we can only derive an explicit comparison for the estimation accuracy under the first kind (only phase-pattern distortion exists). As for the other two kinds, we provide numerical comparisons as an alternative.

For the phase-pattern distortion, we mean that $g_m(\theta) = 1$ and $f_m(\theta) = \exp(j\varphi_m)$. With this condition, Equation (26) should be modified as

$$\mathrm{var}^{(3)}(\hat{\theta}) = \frac{1}{2L\cdot\mathrm{SNR}}\left(1 + \frac{1}{\sum_{m=1}^{M}M\cdot\mathrm{SNR}}\right)\Big/h^{(3)}(\theta) \quad (32)$$

Comparing it with Equation (17), we find that the two functions have the same parameters, with the exception of the dividend $h(\theta)$. Therefore, error variances in *Case 1* and *Case 3* differ only in terms of $h(\theta)$. And $h^{(3)}(\theta)$ is modified as

$$
\begin{aligned}
h^{(3)}(\theta) &= \sum_{m=1}^{M}\left[\omega\tau'_m - \tilde{\varphi}'_m - \sum_{k=1}^{M}\varphi'_k\right]^2 - \frac{1}{M}\left(\sum_{m=1}^{M}\omega\tau'_m - \varphi'_m\right)^2 \\
&= \sum_{m=1}^{M}(\omega\tau'_m - \tilde{\varphi}'_m)^2 - \frac{2}{M}\sum_{m=1}^{M}(\omega\tau'_m - \tilde{\varphi}'_m)\cdot\sum_{k=1}^{M}\varphi'_k + \frac{1}{M}\left(\sum_{m=1}^{M}\varphi'_m\right)^2 \\
&\quad - \frac{1}{M}\left[\left(\sum_{m=1}^{M}\omega\tau'_m\right)^2 - 2\sum_{m=1}^{M}\omega\tau'_m\cdot\sum_{m=1}^{M}\varphi'_m + \left(\sum_{m=1}^{M}\varphi'_m\right)^2\right] \\
&= \sum_{m=1}^{M}(\omega\tau'_m - \tilde{\varphi}'_m)^2 - \frac{1}{M}\left(\sum_{m=1}^{M}\omega\tau'_m\right)^2
\end{aligned}
\tag{33}
$$

where $\tilde{\varphi}'_m = \varphi'_m + 1/M\cdot\sum_{k=1}^{M}\varphi'_k$ and $\sum_{m=1}^{M}\tilde{\varphi}'_m = \sum_{m=1}^{M}\varphi'_m - \sum_{k=1}^{M}\varphi'_k = 0$ are used. In comparison with Equation (16), we find that the error variance in *Case 3* can be smaller than that in *Case 1* under the following condition:

$$
\sum_{m=1}^{M}(\omega\tau'_m - \tilde{\varphi}'_m)^2 > \sum_{m=1}^{M}(\omega\tau'_m)^2
\tag{34}
$$

For numerical comparisons with the other two kinds, we consider a common HFSWR scenario with the following parameters.

- Frequency of operation: $f = 5$ MHz;
- Uniform linear array with eight dipole antennas;
- Antenna spacing: $d = \lambda/2$, with $\lambda$ as the wavelength;
- Monte Carlo number: 2000;
- Assumed antenna pattern distortion:

$$
f_m(\theta) = [0.5 + \cos(\theta + p_m)]\exp[j2\sin(5\theta + q_m)], \qquad m = 2,\cdots,8
\tag{35}
$$

where $p_m$ and $q_m$ follow the uniform distribution of $[-90°, 90°]$. Note that we only consider the third kind (both phase- and gain-pattern distortion exist), since it already includes the second kind (only gain-pattern distortion exists).

Considering a single target source with 5 dB SNR, Figure 1a shows the root mean square error (RMSE) at various incident angles. We employ RMSE instead of EV here, since they are strongly correlated, and RMSE can better represent the estimation performance. It can be found that *Case 3* has a lower RMSE than *Case 1* at all incident angles. Adding an interference source with 0 dB SNR and assuming a 5° angle difference relative to the target source, Figure 1b shows the number of snapshots required to successfully distinguish the two sources. We observe that *Case 3* requires a smaller number of snapshots than *Case 1*. Figure 1a,b demonstrate that employing the measured pattern to estimate DOA can achieve better performance than the ideal situation. We only report the results at an angle of $[-60°, 60°]$ relative to the normal direction of the array, since this is the main area of our interest and is sufficiently representative.

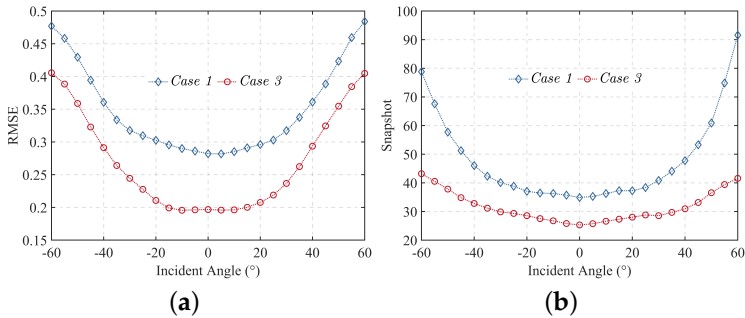

**Figure 1.** (**a**) RMSE and (**b**) the number of snapshots at various incident angles.

## 4. Proposed Calibration Method

In this section, we propose a calibration method for the APD of HFSWR. The method utilizes the first-order sea clutter data and can measure the actual antenna pattern using the artificial hummingbird algorithm. To obtain the available data, we first introduce the extraction methods of first-order sea clutter spectrum and single-DOA spectrum point. Then, we employ the AHA to iteratively estimate the APD.

### 4.1. Extraction of the First-Order Sea Clutter Spectrum

On the Doppler spectrum, first-order sea clutter appears as a continuous spectrum line at the first-order Bragg frequency:

$$f_B = \sqrt{\frac{g}{\pi\lambda}} \tag{36}$$

where $g = 9.8$ m/s$^2$ denotes the gravitational acceleration.

Figure 2 displays a typical Doppler spectrum of the HFSWR. To extract the first-order sea clutter spectrum (take the left spectrum as an example), we first frame it using a sliding window (solid frame in the figure) with a width of $l$ and a center position of $a$. The Doppler resolution is defined as $\delta f$, and the maximum Doppler shift caused by the current velocity is defined as $\Delta f$. Then, the value range of the center position is $a_1 \sim a_2$, where $a_1 = -f_B - \Delta f/2$, and $a_2 = -f_B + \Delta f/2$. The value range of the window's width is $l_1 \sim l_2$, where $l_1 = 2\delta f$, and $l_2 = \Delta f$. Meanwhile, we select a certain area on both sides of the first-order sea clutter window as the noise windows (dashed frames in the Figure 2) with a width of $kl$. The value of $k$ is determined by experience and is usually $0.5 \sim 1.5$.

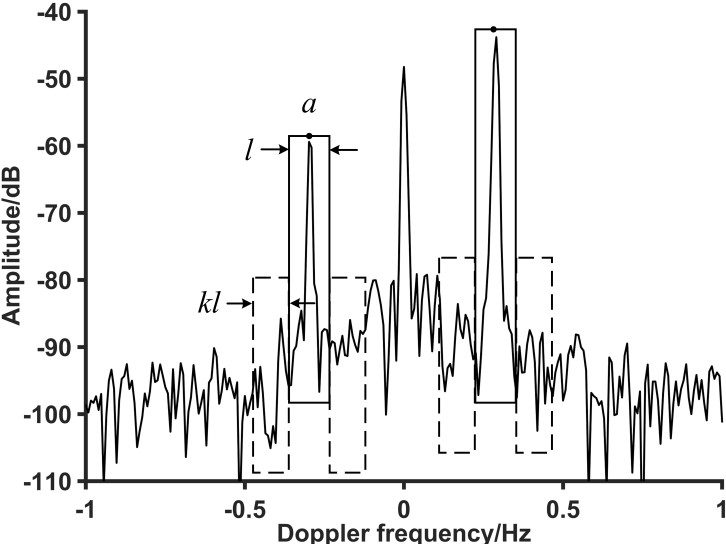

**Figure 2.** The typical Doppler spectrum with sea clutter windows and noise windows. Solid frames are the left and right first-order sea clutter windows. Dashed frames are the corresponding noise windows.

Then, for the left first-order sea clutter spectrum, the sea clutter data inside can be expressed as

$$\text{Clutter}_L = \text{Doppler}(a - 0.5l, a + 0.5l) \tag{37}$$

Note that the noise windows are distributed on both sides of the sea clutter window, so the noise data should be

$$\text{Noise}_L = \text{Doppler}(a - (k + 0.5)l, a + (k + 0.5)l) - \text{Clutter}_L \tag{38}$$

Since the distance between the left and right first-order sea clutter spectral peaks is $2f_B$, we can determine the value range of the center position of the right first-order sea clutter window as $f_B - \Delta f/2 \sim f_B + \Delta f/2$. Then, the sea clutter data and noise data of the right spectrum are

$$\text{Clutter}_\text{R} = \text{Doppler}(a + 2f_B - 0.5l, a + 2f_B + 0.5l) \tag{39}$$

$$\text{Noise}_\text{R} = \text{Doppler}(a + 2f_B - (k + 0.5)l, a + 2f_B + (k + 0.5)l) - \text{Clutter}_\text{R} \tag{40}$$

To ensure the consistency of the extraction results of the left and right first-order sea clutter spectra, we define the SNR as

$$\text{SNR} = \text{SNR}_\text{L} + \text{SNR}_\text{R} \tag{41}$$

where $\text{SNR}_\text{L} = \text{Clutter}_\text{L}/\text{Noise}_\text{L}$ and $\text{SNR}_\text{R} = \text{Clutter}_\text{R}/\text{Noise}_\text{R}$. $a$ and $l$ are slid over their respective value ranges, and their values that maximize SNR are marked as the final values used to determine the first-order sea clutter spectrum.

### 4.2. Extraction of the Single-DOA Spectrum Point

Barrick [31] points out that most of the spectrum points in the first-order sea clutter spectrum are composed of single-source echoes or double-source echoes. This means that we can employ the sea clutter data as the passive calibration source to estimate the actual pattern. To extract the single-DOA spectrum point, a traditional method based on the translation-invariant subarray was proposed in [32]. The example of a ULA shown in Figure 3 explains the translation-invariant subarray. In the Figure 3, subarrays $(1, 2)$ and $(3, 4)$ are a set of translation-invariant subarrays, and subarrays $(1, 3)$ and $(2, 4)$ are also a group of translation-invariant subarrays.

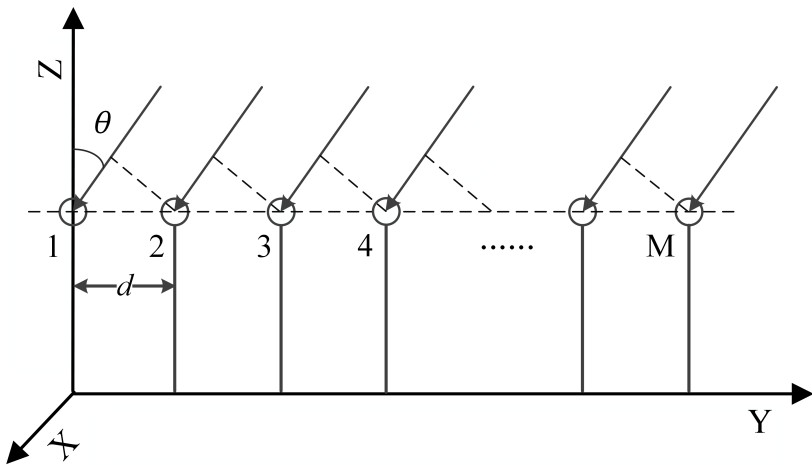

**Figure 3.** Uniform linear array with $M$ antennas.

We now provide a brief account of this method. The method is only applicable to channel mismatch, regardless of the antenna pattern distortion. This indicates that the gain and phase distortions are angle-independent constants. Thus, the received two-dimensional range-Doppler (RD) spectrum of the $m$-th antenna in the ULA is

$$x_m(r, v) = g_m e^{j\varphi_m} \left[ \sum_{n=1}^{N} s_n(r, v) e^{j(m-1)kd \sin \theta_n} \right] + n_m(r, v) \tag{42}$$

where

- $g_m$ and $\varphi_m$ denote the gain- and phase-distortion constants, respectively, of the $m$-th channel;

- $r$ and $v$ represent the range and Doppler coordinates of the RD spectrum, respectively;
- $k = 2\pi/\lambda$, with $\lambda$ as the wavelength of the radar signal.

Based on the translation invariance of subarrays $(1,2)$ and $(3,4)$, a variate $\eta_1$ is defined as

$$\eta_1 = \frac{x_1(r,v)x_4(r,v)}{x_2(r,v)x_3(r,v)} \tag{43}$$

If the spectrum point is a single-source echo and the noise is ignored, that is, $N = 1$ and $n_m(r,v) = 0$, then $\eta_1$ is reduced to

$$\eta_1 = \frac{g_1 g_4}{g_2 g_3} \exp[j(\varphi_1 + \varphi_4 - \varphi_2 - \varphi_3)] \tag{44}$$

It is observed that $\eta_1$ is a constant independent of the incident angle of the source. Of course, in the actual application, considering the noise's effect, $\eta_1$ will be clustered around this constant with a slight deviation. On the contrary, for the multiple-source case, i.e., $N \geq 2$, $\eta_1$ varies with incident angle and diverges throughout the complex plane. According to the convergence property of $\eta_1$, the single-DOA spectrum points clustered in the complex plane are identified and represented by set $Q_1$.

To improve the accuracy of extraction, multiple variates can be defined similar to $\eta_1$:

$$\eta_2 = \frac{x_1(r,v)x_2^{\mathrm{H}}(r,v)}{x_3(r,v)x_4^{\mathrm{H}}(r,v)} \qquad \eta_3 = \frac{x_1(r,v)x_3^{\mathrm{H}}(r,v)}{x_2(r,v)x_4^{\mathrm{H}}(r,v)} \tag{45}$$

The convergent points of $\eta_2$ and $\eta_3$ in the complex plane are still selected as the single-DOA spectrum points and represented by sets $Q_2$ and $Q_3$, respectively. Since the probability of multiple-DOA spectrum points falling into three clustering areas at the same time is extremely low, the intersection of $Q_1$, $Q_2$ and $Q_3$ is chosen as the final single-source spectral point set.

However, when considering the APD, the received two-dimensional RD spectrum of the $m$-th antenna should be

$$x_m(r,v) = \left[ \sum_{n=1}^{N} g_m(\theta_n)e^{j\varphi_m(\theta_n)}s_n(r,v)e^{j(m-1)kd\sin\theta_n} \right] + n_m(r,v) \tag{46}$$

Then, the variate $\eta_1$ should be modified as

$$\eta_1 = \frac{g_1(\theta)g_4(\theta)}{g_2(\theta)g_3(\theta)} \exp[j(\varphi_1(\theta) + \varphi_4(\theta) - \varphi_2(\theta) - \varphi_3(\theta))] \tag{47}$$

At this time, $\eta_1$ is a variable related to the incident angle of the source, so the traditional extraction method is unavailable. To overcome this deficiency, we combine the traditional extraction method with the beamforming algorithm and extract the single-DOA spectrum point sector by sector. We divide the airspace into several sectors whose sizes are equal to the half-power beam width (HPBW) of the array pattern, as shown in Figure 4. We consider that the APD within a sector varies less and, further, assume that it is a constant equal to the APD at the sector's center angle. Then, the specific extraction steps are as follows.

*Step 1:* Select a concrete sector and perform beamforming on the channel RD spectra to obtain the beam RD spectrum. Note that the Chebyshev beamforming algorithm is employed, and the beamforming points to the center angle of the selected sector.

*Step 2:* Select the first-order sea clutter points with high SNR (typically $> 20$ dB) in the beam RD spectrum and record their coordinate set ($\{(r_i, v_i); q = 1, 2, \cdots, Q\}$).

*Step 3:* Return to the channel RD spectra and use the traditional extraction method to extract the single-DOA spectrum points from the coordinate set recorded in *Step 2*. The finalized spectrum points are regarded as the single-DOA spectrum points within this sector.

*Step 4:* Change sectors and repeat *steps 1~3* until all sectors are processed.

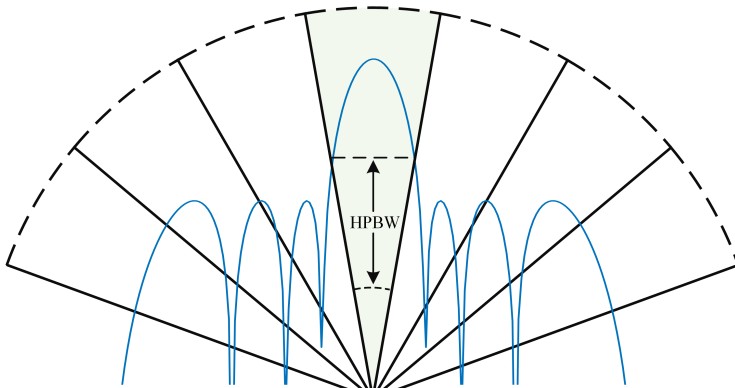

**Figure 4.** Airspace diagram of the sectors.

*4.3. Iterative Estimation of the Antenna Pattern Distortion*

Note that the single-DOA spectrum points extracted in Section 4.2 are only meaningful in their respective sectors, so estimation of the APD should also be performed within the sector. Since airspace information is used for sector division, we implement APD estimation within the sector by employing the Beamspace MUSIC algorithm, which uses beamspace information to estimate DOA. The BMUSIC algorithm utilizes the beamspace transformation matrix to convert the element-space data ($\mathbf{X}(t)$) into beamspace data $\mathbf{Y}(t)$:

$$\mathbf{Y}(t) = \mathbf{T}^H \mathbf{X}_{\text{APD}}(t) = \mathbf{T}^H \mathbf{A}_{\text{APD}}(\theta)\mathbf{S}(t) + \mathbf{N}_{\text{B}}(t) \tag{48}$$

where $\mathbf{T}$ is the beamspace transformation matrix, superscript H denotes the Hermitian transpose and $\mathbf{N}_{\text{B}}(t) = \mathbf{T}^H \mathbf{N}(t)$ represents the beamspace noise. The transformation matrix ($\mathbf{T}$) satisfies $\mathbf{T}^H \mathbf{T} = \mathbf{I}$; its exact composition is covered later in this paper.

Then, the covariance matrix of the beamspace data is

$$\begin{aligned} \mathbf{R}_{\text{Y}} &= E\left[\mathbf{Y}(t)\mathbf{Y}^H(t)\right] = \mathbf{T}^H E\left[\mathbf{X}_{\text{APD}}(t)\mathbf{X}_{\text{APD}}^H(t)\right]\mathbf{T} \\ &= \mathbf{T}^H \mathbf{A}_{\text{APD}} \mathbf{R}_{\text{S}} \mathbf{A}_{\text{APD}}^H \mathbf{T} + \sigma^2 \mathbf{T}^H \mathbf{T} \\ &= \mathbf{B} \mathbf{R}_{\text{S}} \mathbf{B}^H + \sigma^2 \mathbf{I} \end{aligned} \tag{49}$$

where $\mathbf{B} = \mathbf{T}^H \mathbf{A}_{\text{APD}} = \left[\mathbf{T}^H \mathbf{f}(\theta_1) \odot \mathbf{a}(\theta_1), \mathbf{T}^H \mathbf{f}(\theta_2) \odot \mathbf{a}(\theta_2), \cdots, \mathbf{T}^H \mathbf{f}(\theta_N) \odot \mathbf{a}(\theta_N)\right] = [\mathbf{b}(\theta_1),$ $\mathbf{b}(\theta_2), \cdots, \mathbf{b}(\theta_N)]$ denotes the array manifold of the beam space.

According to the principle of MUSIC, the spectrum function of the BMUSIC algorithm is

$$P_{\text{B}}(\theta) = \mathbf{b}^H(\theta)\tilde{\mathbf{U}}_{\text{N}}\tilde{\mathbf{U}}_{\text{N}}^H \mathbf{b}(\theta) \tag{50}$$

where $\tilde{\mathbf{U}}_{\text{N}}$ denotes the noise subspace of beamspace covariance matrix $\mathbf{R}_{\text{Y}}$.

A novel method is proposed to iteratively estimate the APD and DOA based on BMUSIC and the artificial hummingbird algorithm (AHA). The steps included in the method are described as follows.

*Initialization:* Divide the air space into several sectors in the same way as described Section 4.2 and set the sector counter and iteration counter to $s = 1$ and $i = 0$, respectively. Then, initialize the antenna patterns to ideal values, that is, $f_m^{(i)}(\theta) = f_m^{(0)}(\theta) = f_1(\theta) = 1$, where $m = 2, \cdots, M$.

*Step 1:* Select the set of single-DOA spectrum points of the $s$-th sector and fix the APD vector as $\mathbf{f}^{(i)}(\theta) = [f_1^{(i)}(\theta), \cdots, f_M^{(i)}(\theta)]^T$. For each spectrum point ($\mathbf{X}_q(t)(q = 1, \cdots, Q)$), convert it to the beamspace spectrum point ($\mathbf{Y}_q(t)(q = 1, \cdots, Q)$) according to Equation (48). The beamspace transformation matrix ($\mathbf{T}$) is determined based on the selected sector using the Chebyshev beamforming algorithm.

$$\mathbf{T} = \mathbf{W} \odot [\mathbf{f}(\theta_{\text{start}}) \odot \mathbf{a}(\theta_{\text{start}}), \mathbf{f}(\theta_{\text{start}} + \Delta\theta) \odot \mathbf{a}(\theta_{\text{start}} + \Delta\theta), \cdots, \mathbf{f}(\theta_{\text{end}}) \odot \mathbf{a}(\theta_{\text{end}})] \tag{51}$$

where

- **W** denotes the Chebyshev coefficient vector;
- $\theta_{\text{start}}$ and $\theta_{\text{end}}$ represent the start and end angles of the *s*-th sector, respectively;
- $\Delta\theta = (\theta_{\text{end}} - \theta_{\text{start}})/(B-1)$, where $B$ denotes the number of beams and is generally equal to the number of antennas, i.e., $B = M$.

Note that **T** in Equation (51) does not satisfy the orthogonality condition ($\mathbf{T}^{\text{H}}\mathbf{T} = \mathbf{I}$), so the final beamspace transformation matrix (**T**) should be modified by orthogonal processing:

$$\mathbf{T} = \mathbf{T}(\mathbf{T}^{\text{H}}\mathbf{T})^{-1/2} \tag{52}$$

*Step 2:* For each beamspace spectrum point ($\mathbf{Y}_q(t)$), estimate the DOA $\theta_q^{(i)}$ by minimizing the spectrum function described in Equation (50) and construct the estimated DOA set ($\{\theta_q^{(i)}\}_{q=1}^Q$) and array manifold set ($\{\mathbf{a}(\theta_q^{(i)})\}_{q=1}^Q$).

*Step 3:* Construct the cost function based on the BMUSIC algorithm:

$$J = \sum_{q=1}^Q \mathbf{b}^{\text{H}}(\theta_q)\tilde{\mathbf{U}}_{\text{N}}\tilde{\mathbf{U}}_{\text{N}}^{\text{H}}\mathbf{b}(\theta_q) = \sum_{q=1}^Q \mathbf{a}^{\text{H}}(\theta_q) \odot \mathbf{f}^{\text{H}}(\theta_q)\mathbf{T}\tilde{\mathbf{U}}_{\text{N}}\tilde{\mathbf{U}}_{\text{N}}^{\text{H}}\mathbf{T}^{\text{H}}\mathbf{f}(\theta_q) \odot \mathbf{a}(\theta_q) \tag{53}$$

The cost function obtains the minimum value when the estimated ADPs and DOAs are the same as their true values. Therefore, the problem can be regarded as the minimum optimization problem of the cost function ($J$) with the fixed array manifold set ($\{\mathbf{a}(\theta_q^{(i)})\}_{q=1}^Q$).

Then, employ the artificial hummingbird algorithm [33] to solve this optimization problem and obtain the estimated APD vector set ($\{\mathbf{f}_e(\theta_q)\}_{q=1}^Q$). Since the APD includes the gain distortion and phase distortion, i.e., $f(\theta) = g(\theta)\exp(j\varphi(\theta))$, the AHA should be initialized to $2M$ dimensions. The first $M$ dimensions and the last $M$ dimensions represent the gain distortion and phase distortion of the $M$ antennas, respectively.

*Step 4:* For each interested angle within the *s*-th sector, update the APD vector based on the following principle:

$$\mathbf{f}^{(i+1)}(\theta) = \begin{cases} \mu\mathbf{f}^{(i)}(\theta) + (1-\mu)\mathbf{f}_e(\theta) & \text{if } \theta \in \{\theta_q^{(i)}\}_{q=1}^Q \\ \mathbf{f}^{(i)}(\theta) & \text{if } \theta \in \text{others} \end{cases} \tag{54}$$

where $\mu$ is determined based on experience. Then, utilize $\mathbf{f}^{(i+1)}(\theta)$ instead of $\mathbf{f}^{(i)}(\theta)$ in *Step 1* and repeat *Step 2* to update the estimated DOA set: $\{\theta_q^{(i+1)}\}_{q=1}^Q$.

*Step 5:* Set the maximum iterations ($I$) and iteration termination threshold ($\varepsilon$), which usually has a minimum value greater than zero. Substitute $\mathbf{f}^{(i+1)}(\theta)$ and $\{\theta_q^{(i+1)}\}_{q=1}^Q$ into Equation (53) to calculate the updated cost function value ($J^{(i+1)}$). Repeat *Steps 2~4* to iteratively estimate the APD vector, unless $i = I$ or

$$J^{(i)} - J^{(i+1)} < \varepsilon \tag{55}$$

*Step 6:* Let $s = s + 1$ and return to *Step 1* until all sectors have been processed.

## 5. Numerical and Experimental Results

In this section, we perform numerical simulations and experiments to validate the reliability of our proposed method.

### 5.1. Numerical Results

In the numerical simulation, we presume the APD vector of the array and simulate the first-order sea clutters according to the method proposed in [34]. A specific with the following parameters is considered.

- Radar operating frequency: $f = 8.15$ MHz;
- Airspace range: $[-90°, 90°]$ relative to the normal direction of the array;
- The array configuration is the same as in Figure 3;
- Number of dipole antennas: $M = 4$, for simplicity;
- Antenna spacing: $d = 14.5$ m;
- APDs of different antennas (Figure 5) are generated according to the following criteria:

  * Set the first antenna as the reference antenna;
  * Divide the range of interest into nine sectors, with each APD controlled by the midpoints of these sectors;
  * Each midpoint contains gain and phase distortion, with the gain distortion and phase distortion obeying a uniform distribution in the range of $[0.5, 1.5]$ and $[-1, 1]$ (radians), respectively;
  * Each APD curve is obtained by these midpoints through cubic spline interpolation.

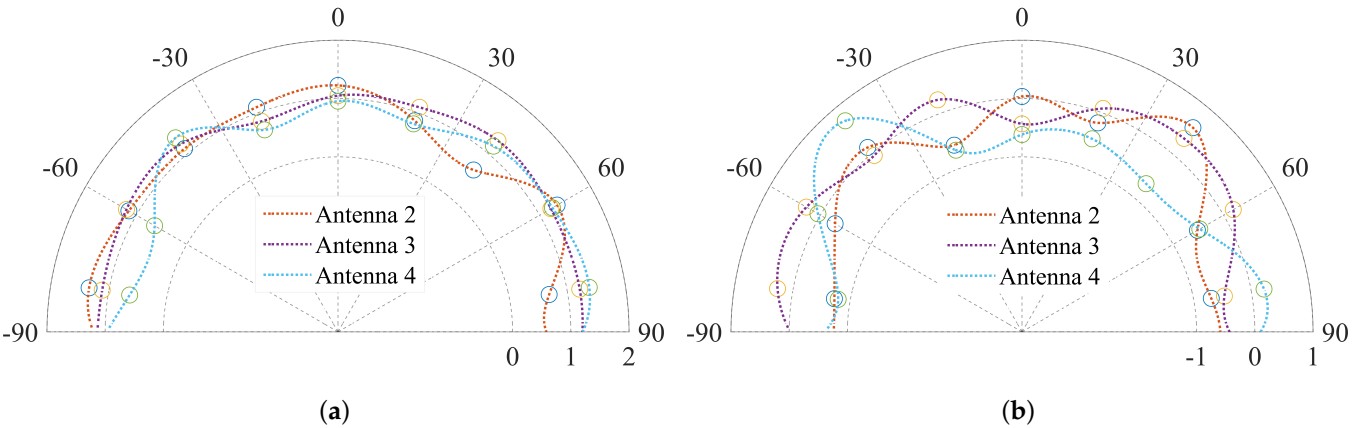

**Figure 5.** Presumed (**a**) gain and (**b**) phase distortion of the antennas.

We first compare the DOA estimation performance of the ideal and actual patterns to further verify the theory proposed in Section 3. The signal is incident from 30° relative to the normal direction of the array, with half-wavelength antenna spacing. In Figure 6a, we illustrate the root mean square error (RMSE) and probability of detection (PD) under various SNRs, with a fixed snapshot of 64. By successful detection, we mean that the DOA estimate error is less than 2°. We carry out 500 Monte Carlo simulations and observe that compared with the ideal pattern, the actual pattern can significantly decrease the RMSE and increase the PD of DOA estimation, especially under low SNR values. The RMSE and PD with different snapshots are shown in Figure 6b. We employ a 5 dB SNR and maintain other parameters here. Similarly, it is found that using the actual pattern can lead to better DOA estimation performance than the ideal pattern, especially under low snapshots, which proves our conclusion proposed in Section 3.

We then evaluate the calibration method we proposed in Section 4 according to the simulated sea clutter. We compare the estimated APD and actual APD and show the results in Figures 7 and 8. As can be seen from the two figures, the estimated and actual gain distortion have a good coincidence for each antenna, as well as the estimated and actual phase distortion. Note that the sectors near the normal direction (0°) have better consistency than the sectors far away from the normal direction, which is due to the inherent properties of the MUSIC algorithm. In the angle range of −30° to 30°, the estimated and actual gain distortion are basically consistent. Meanwhile, affected by the Vandermonde characteristic of the array manifold of the ULA, the estimation effect of this calibration method on phase distortion is worse than that on gain distortion.

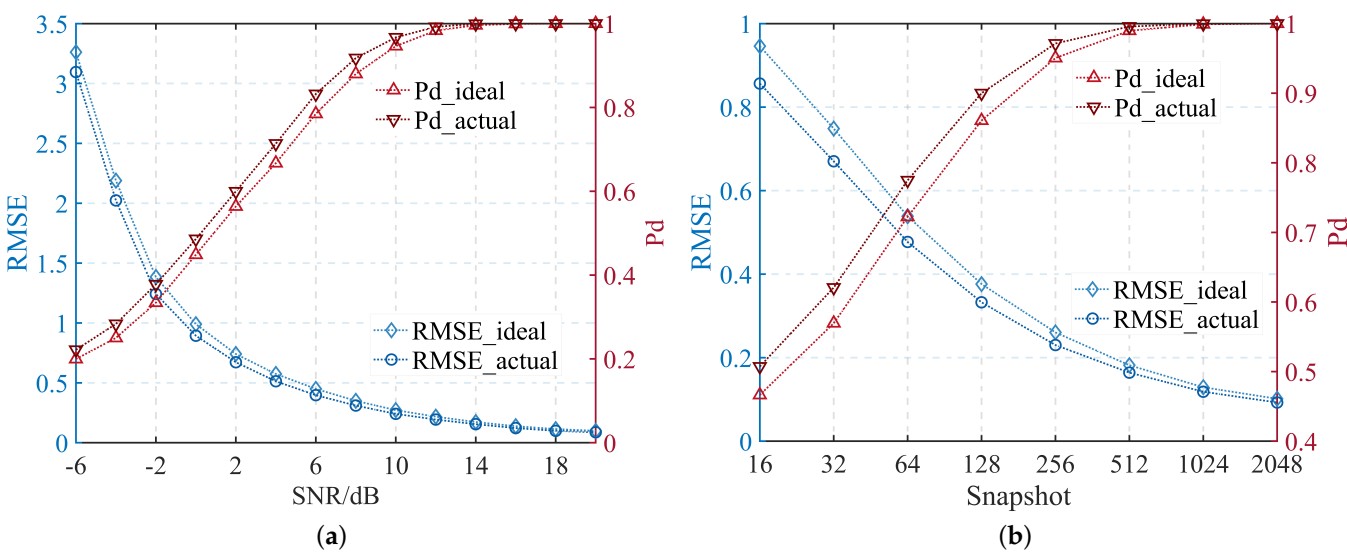

**Figure 6.** RMSE and PD of DOA estimation under various (**a**) SNRs and (**b**) snapshots.

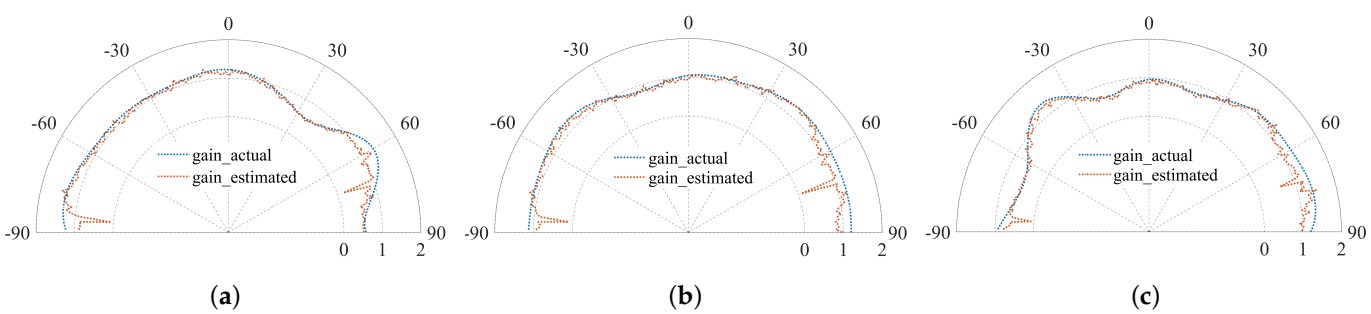

**Figure 7.** Comparison of estimated and actual gain distortions of (**a**) antenna 2, (**b**) antenna 3 and (**c**) antenna 4.

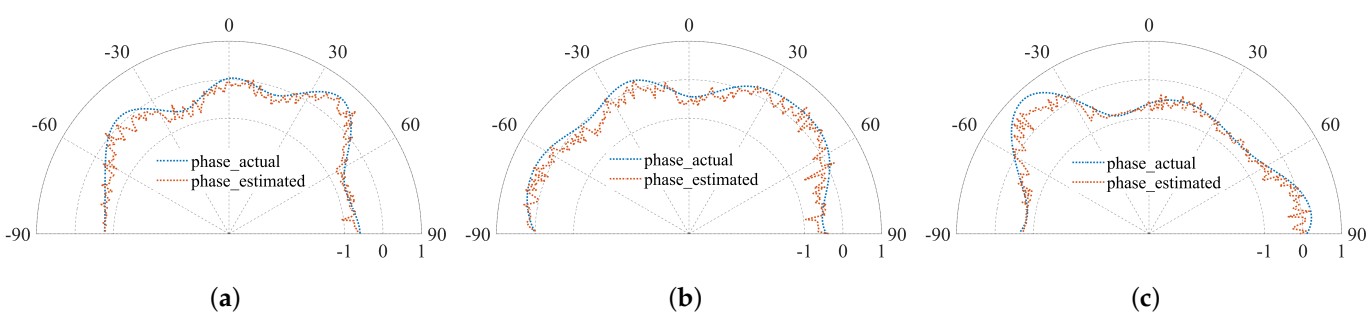

**Figure 8.** Comparison of estimated and actual phase distortions of (**a**) antenna 2, (**b**) antenna 3 and (**c**) antenna 4.

### 5.2. Experimental Results

To further verify the feasibility of the proposed method, we estimate the APD based on the actual data measured by the Weihai HFSWR site. The site is shown in Figure 9; the radar operating frequency is 8.15 MHz, the receiving array is a uniform linear array of 8 elements and the antenna spacing is 14.5 m.

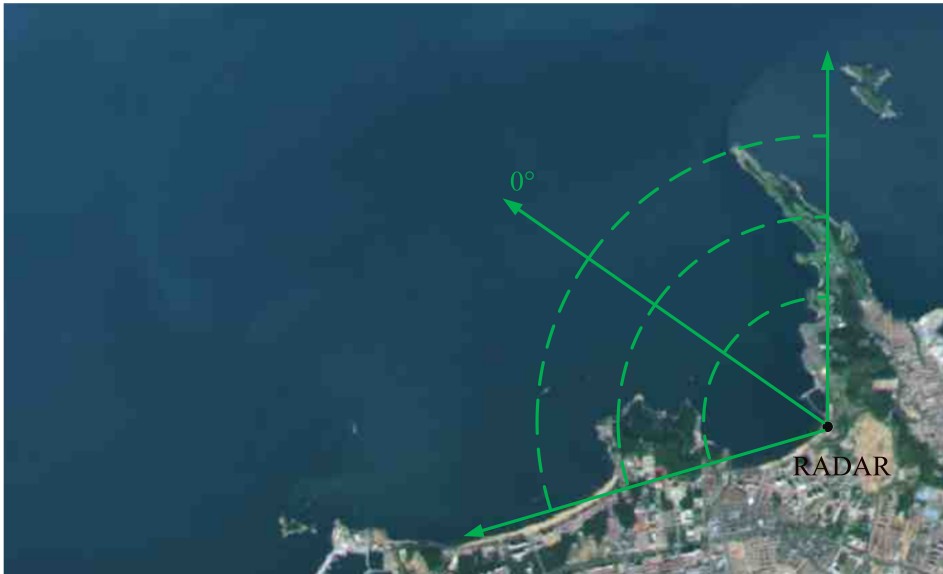

**Figure 9.** Experimental site.

Note that due to the limitations of the terrain, only an airspace range of only $[-60°, 60°]$ is of concern. We divide this range into six sectors at intervals of $20°$ and show the RD spectrum of the sector centered at $-10°$ in Figure 10a, in which two distinct first-order sea clutter spectra can be seen. Figure 10b illustrates the sea clutter spectral extraction results of the 50th range index in the RD spectrum shown in Figure 10a. The red and black dotted lines represent the sea clutter and noise windows, respectively. The range of sea clutter is framed by the red line, which proves the correctness of our extraction method. Note that due to the effect of wind speed, the amplitudes of the left and right first-order spectra are different. We also present the clustering results of $\eta_1 \sim \eta_3$ in this sector in Figure 11. In the figure, the neighborhood radius is set to 0.5, and the red and blue dots represent the convergence and divergence points, respectively. The single-DOA spectrum points are determined by the intersection of these three clustering results.

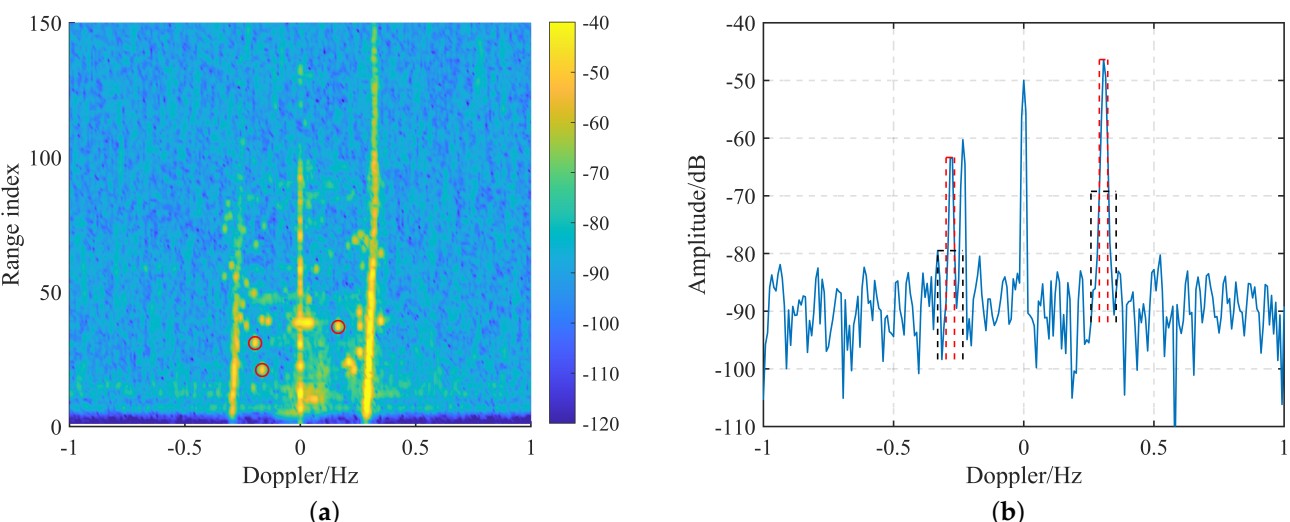

**Figure 10.** (**a**) RD spectrum of the sector centered at $-10°$. (**b**) First-order sea clutter spectral extraction results of the 50th range index.

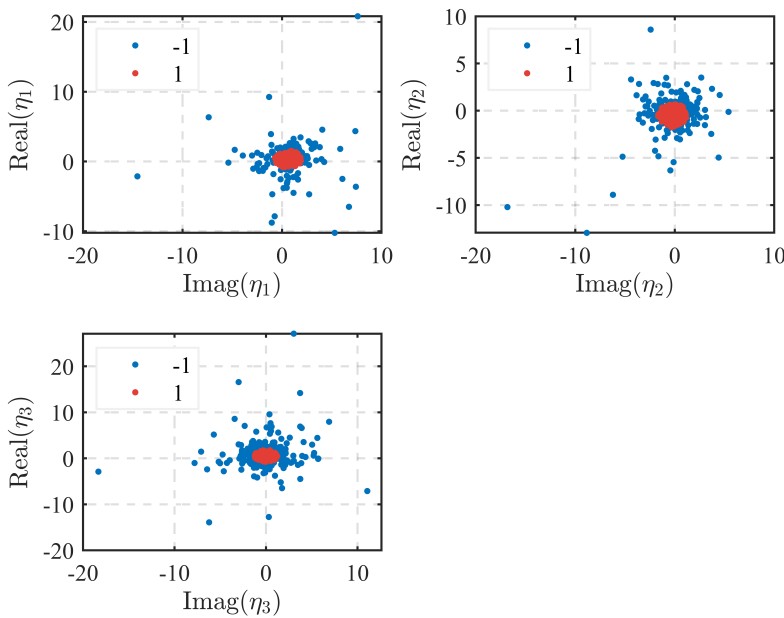

**Figure 11.** Clustering results of variables $\eta_1$, $\eta_2$ and $\eta_3$.

Based on the single-DOA spectrum points, we estimate the APDs of the antennas and illustrate the results in Figure 12. For clarity of the figure, we only show the APDs of the second to fourth antennas. Note that we only report the range of interest. It can be found that each antenna has a certain degree of antenna pattern distortion. However, since we cannot measure the actual pattern (which is quite costly), it is impossible to evaluate the reliability of the proposed method by comparing actual and estimated APD. As an alternative, we select several ship echoes, which are marked in Figure 10a with red circles, and obtain their actual directions by employing the AIS data. We estimate their DOAs using the estimated APD and the ideal pattern, respectively. The results are plotted in Figure 13. We observe that in various directions, using the estimated APD can achieve better direction-finding performance than using the ideal pattern, including higher spectral peaks and lower estimation errors. By taking the mean, we calculate that the amplitude improvement and accuracy improvement are about 10 dB and 2°, respectively. This means that our estimated APD is relatively close to the actual value and, to a certain extent, proves the correctness of the calibration method we proposed.

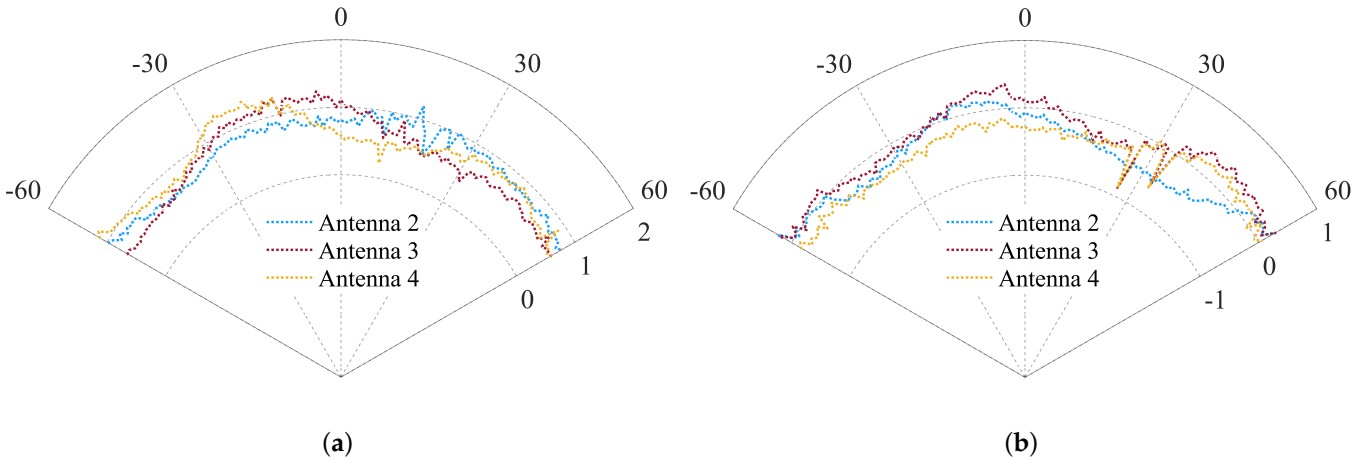

(**a**)                                                                                        (**b**)

**Figure 12.** Estimated (**a**) gain distortions and (**b**) phase distortions of the antennas.

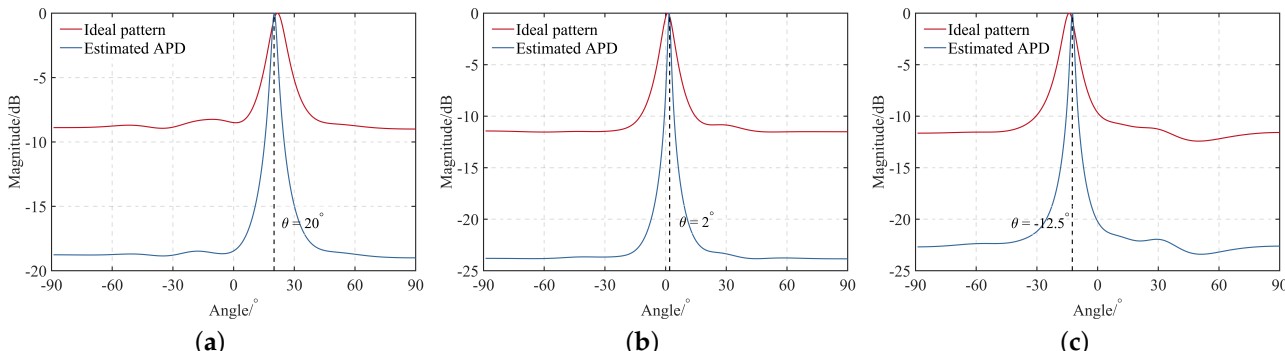

**Figure 13.** Spatial spectrum under the ideal pattern and the estimated APD of (**a**) 20°, (**b**) 2° and (**c**) −12.5° ship echoes.

## 6. Conclusions

In this paper, we first performed a detailed analysis of MUSIC performance in terms of the estimation accuracy and angular resolution. We derived their explicit expressions under the ideal pattern and APD cases. By employing the theoretical and numerical analyses, we demonstrated that using the actual pattern can improve the direction-finding performance. Based on this proposition, we proposed an iterative calibration method for the APD of HFSWR, which can realize the joint estimation of DOA and APD. We utilized the first-order sea clutter data as the calibration source and proposed extraction methods of sea clutter spectrum and single-DOA spectrum points. In each iteration, the BMUSIC algorithm and AHA were used to estimate the DOA and APD of the single-DOA point, respectively. Numerical results show a good coincidence between the actual pattern and the estimated APD. In particular, within the angle range of −30° to 30°, the estimated results and the actual pattern are basically consistent. Meanwhile, we further proved that the actual pattern outperforms the ideal pattern in terms of DOA estimation. We experimentally obtained the APDs of the HFSWR antennas and improved the direction-finding performance of several actual ship targets based on the obtained APDs. We obtained an amplitude improvement of approximately 10 dB and a 2° accuracy improvement in the spatial spectrum. Both numerical and experimental results prove the correctness of our proposed calibration method.

**Author Contributions:** Conceptualization, H.L., A.L. and Q.Y.; methodology, H.L.; software, H.L.; validation, H.L.; formal analysis, H.L., C.Y. and Z.L.; investigation, Z.L.; resources, Z.L.; data curation, H.L.; writing—original draft preparation, H.L.; writing—review and editing, H.L. and A.L.; visualization, H.L.; supervision, H.L.; project administration, H.L.; funding acquisition, A.L. and C.Y. All authors have read and agreed to the published version of the manuscript.

**Funding:** This research was funded by the National Nature Science Foundation of China under Grants 62031015, 61971159 and 61901137.

**Data Availability Statement:** The data are not accessible to the public. Please contact the corresponding author for data.

**Acknowledgments:** We are grateful to the editor and anonymous reviewers for their suggestions. We thank the researchers at the radar station of the Harbin Institute of Technology, Weihai, for providing us with high-frequency surface wave radar data.

**Conflicts of Interest:** The authors declare no conflict of interest.

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
