# Peer review of "Antenna Pattern Calibration Method for Phased Array of High-Frequency Surface Wave Radar Based on First-Order Sea Clutter"

_remotesensing, doi:10.3390/rs15245789_

Round 1

Reviewer 1 Report

Comments and Suggestions for Authors

The manuscript by Li et al. entitled “Antenna pattern calibration method for phased array of high-frequency surface wave radar based on first-order sea clutter,” as the title implies, investigates the effect of antenna pattern distortions on backscatter data obtained from shore based high frequency (HF) radar systems. This topic has been addressed by other authors but this work is novel in that: 1) it looks explicitly at the effect of antenna pattern distortions on phased array systems, which has been ignored to date, 2) it provides a theoretical derivation of the distortion effects, and 3) it presents a self-contained method to estimate antenna pattern distortions from the collected backscatter data.

The use of shore-based HF radar systems for marine science, military, and coastal resource management applications has grown in recent years. Furthermore, the users of these systems are most often environmental scientists without the in-depth background in signal processing required to fully understand or to minimize errors associated with the measurements. This author team represents an exception to that rule. They have been conducting detailed analyses of backscatter data and detailed theoretical derivations based on the complicated signal processing algorithms typically employed by the HF radar systems. Others have done so as well and the authors here do a good job reviewing and referencing past work.

The manuscript is clearly written and illustrated with only a few minor exception as noted below. The validation results presented using numerical simulations and collected backscatter data are convincing and should be of interest to users of HF radar systems. For these reasons, I recommend the manuscript for publication with only minor corrections.

MINOR COMMENTS:

Line 8: “method which” should be “method, which”

Line 46: “mature since their” should be “mature due to their”

Line 46: “models which” should be “models, which”

Line 79: “through the iterative way” should be “through iteration”

Line 131: “Since its” should be “Due to its”

Line 294: “means that employ” should be “this means that we can employ”

Line 362: “steps the” should be “steps in the”

Reviewer 2 Report

Comments and Suggestions for Authors

Antenna pattern distortion will decrease the direction-of-arrival (DOA) estimation performance of the direction finding algorithm using the phase manifold of HF radar. The authors propose a method for array calibration from first-order sea clutter spectrum and single-DOA spectrum points. The antenna pattern calibration has been focused for Direction-finding HF radars over long years, while the phase distortion has been found recently for phased-array HF radar, which is focused in this work. The paper is well written and is a good report for the HF radar community. Some minor comments for the paper. 

1. line 269, this is not range-Doppler spectrum. It is Doppler spectrum.

2. line 271-272, “affected by … spectrum broadening”. I’m confused. To my knowledge, spectrum broadening is complicated.

3. In fig 8 and 12, it is strange for the phase_ideal line, please give more details on it.

4. part 3 can be found in other literature, I suggest that this part can be cited.

Comments on the Quality of English Language

Minor editing of English language required.

Reviewer 3 Report

Comments and Suggestions for Authors

Round 2

Reviewer 3 Report

Comments and Suggestions for Authors

I congratulate the authors to provide a detailed rebuttal to the major comments. Then, the paper is OK to be published.